# SIMPLE AND EFFICIENT ARCHITECTURE SEARCH FOR CONVOLUTIONAL NEURAL NETWORKS

## ABSTRACT

Neural networks have recently had a lot of success for many tasks. However, neural network architectures that perform well are still typically designed manually by experts in a cumbersome trial-and-error process. We propose a new method to automatically search for well-performing CNN architectures based on a simple hill climbing procedure whose operators apply network morphisms, followed by short optimization runs by cosine annealing. Surprisingly, this simple method yields competitive results, despite only requiring resources in the same order of magnitude as training a single network. E.g., on CIFAR-10, our method designs and trains networks with an error rate below 6% in only 12 hours on a single GPU; training for one day reduces this error further, to almost 5%.

## 1 INTRODUCTION

Neural networks have rapidly gained popularity over the last few years due to their success in a variety of tasks, such as image recognition (Krizhevsky et al., 2012), speech recognition (Hinton et al., 2012) and machine translation (Bahdanau et al., 2015). In most cases, these neural networks are still designed by hand, which is an exhausting, time-consuming process. Additionally, the vast amount of possible configurations requires expert knowledge to restrict the search. Therefore, a natural goal is to design optimization algorithms that automate this neural architecture search.

However, most classic optimization algorithms do not apply to this problem, since the architecture search space is discrete (e.g., number of layers, layer types) and conditional (e.g., the number of parameters defining a layer depends on the layer type). Thus, methods that rely on, e.g., differentiability or independent parameters are not applicable. This led to a growing interest in using evolutionary algorithms (Real et al., 2017; Suganuma et al., 2017) and reinforcement learning (Baker et al., 2016; Cai et al., 2017; Zoph & Le, 2017) for automatically designing CNN architectures. Unfortunately, most proposed methods are either very costly (requiring hundreds or thousands of GPU days) or yield non-competitive performance.

In this work, we aim to dramatically reduce these computational costs while still achieving competitive performance. Specifically, our contributions are as follows:

- We propose a baseline method that *randomly* constructs networks and trains them with SGDR (Loshchilov & Hutter, 2017). We demonstrate that this simple baseline achieves 6%-7% test error on CIFAR-10, which already rivals several existing methods for neural archictecture search. Due to its simplicity, we hope that this baseline provides a valuable starting point for the development of more sophisticated methods in the future.

- We formalize and extend the work on network morphisms (Chen et al., 2015; Wei et al., 2016; Cai et al., 2017) in order to provide popular network building blocks, such as skip connections and batch normalization.

- We propose Neural Architecture Search by Hillclimbing (NASH), a simple iterative approach that, at each step, applies a set of alternative network morphisms to the current network, trains the resulting child networks with short optimization runs of cosine annealing (Loshchilov & Hutter, 2017), and moves to the most promising child network. NASH finds and trains competitive architectures at a computational cost of the same order of magnitude as training a single network; e.g., on CIFAR-10, NASH finds and trains CNNs with an error rate below 6 % in roughly 12 hours on

a single GPU. After one day the error is reduced to almost 5%. Models from different stages of our algorithm can be combined to achieve an error of 4.7 % within two days on a single GPU. On CIFAR-100, we achieve an error below 24% in one day and get close to 20% after two days.

- Our method is easy to use and easy to extend, so it hopefully can serve as a basis for future work.

We first discuss related work in Section 2. Then, we formalize the concept of network morphisms in Section 3 and propose our architecture search methods based on them in Section 4. We evaluate our methods in Section 5 and conclude in Section 6.

## 2 RELATED WORK

**Hyperparameter optimization.** Neural networks are known to be quite sensitive to the setting of various hyperparameters, such as learning rates and regularization constants. There exists a long line of research on automated methods for setting these hyperparameters, including, e.g., random search (Bergstra & Bengio, 2012), Bayesian optimization (Bergstra et al., 2011; Snoek et al., 2012), bandit-based approaches (Li et al., 2016a), and evolutionary strategies (Loshchilov & Hutter, 2016).

**Automated architecture search.** In recent years, the research focus has shifted from optimizing hyperparameters to optimizing architectures. While architectural choices can be treated as categorical hyperparameters and be optimized with standard hyperparameter optimization methods (Bergstra et al., 2011; Mendoza et al., 2016), the current focus is on the development of special techniques for architectural optimization. One very popular approach is to train a reinforcement learning agent with the objective of designing well-performing convolutional neural networks (Baker et al., 2016; Zoph & Le, 2017; Cai et al., 2017). Baker et al. (2016) train an RL agent to sequentially choose the type of layers (convolution, pooling, fully connected) and their parameters. Zoph & Le (2017) use a recurrent neural network controller to sequentially generate a string representing the network architecture. Both approaches train their generated networks from scratch and evaluate their performance on a validation set, which represents a very costly step. In a follow-up work (Zoph et al., 2017), the RL agent learned to build cells, which are then used as building blocks for a neural network with a fixed global structure. Unfortunately, training an RL agent with the objective of designing architecture is extremely expensive: both Baker et al. (2016) and Zoph & Le (2017) required over 10.000 fully trained networks, requiring hundreds to thousands of GPU days. To overcome this drawback, Cai et al. (2017) proposed to apply the concept of network transformations/morphisms within RL. As in our (independent, parallel) work, the basic idea is to use the these transformation to generate new pre-trained architectures to avoid the large cost of training all networks from scratch. Compared to this work, our approach is much simpler and 15 times faster while obtaining better performance.

Real et al. (2017) and Suganuma et al. (2017) utilized evolutionary algorithms to iteratively generate powerful networks from a small network. Operations like inserting a layer, modifying the parameters of a layer or adding skip connections serve as "mutations" in their framework of evolution. Whereas Real et al. (2017) also used enormous computational resources (250 GPUs, 10 days), Suganuma et al. (2017) were restricted to relatively small networks due to handling a population of networks. In contrast to the previous methods where network capacity increases over time, Saxena & Verbeek (2016) start with training a large network (a "convolution neural fabric") and prune this in the end. Very recently, Brock et al. (2017) used hypernetworks (Ha et al., 2017) to generate the weights for a randomly sampled network architecture with the goal of eliminating the costly process of training a vast amount of networks.

**Network morphism/ transformation.** Network transformations were (to our knowledge) first introduced by Chen et al. (2015) in the context of transfer learning. The authors described a function preserving operation to make a network deeper (dubbed "Net2Deeper") or wider ("Net2Wider") with the goal of speeding up training and exploring network architectures. Wei et al. (2016) proposed additional operations, e.g., for handling non-idempotent activation functions or altering the kernel size and introduced the term network morphism. As mentioned above, Cai et al. (2017) used network morphisms for architecture search, though they just employ the Net2Deeper and Net2Wider operators from Chen et al. (2015) as well as altering the kernel size, i.e., they limit their search space to simple architectures without, e.g., skip connections.

## 3 NETWORK MORPHISM

Let $\mathcal{N}(\mathcal{X})$ denote a set of neural networks defined on $\mathcal{X} \subset \mathbb{R}^n$. A network morphism is a mapping $M : \mathcal{N}(\mathcal{X}) \times \mathbb{R}^k \to \mathcal{N}(\mathcal{X}) \times \mathbb{R}^j$ from a neural network $f^w \in \mathcal{N}(\mathcal{X})$ with parameters $w \in \mathbb{R}^k$ to another neural network $g^{\tilde{w}} \in \mathcal{N}(\mathcal{X})$ with parameters $\tilde{w} \in \mathbb{R}^j$ so that

$$f^w(x) = g^{\tilde{w}}(x) \quad \text{for every } x \in \mathcal{X}. \tag{1}$$

In the following we give a few examples of network morphisms and how standard operations for building neural networks (e.g., adding a convolutional layer) can be expressed as a network morphism. For this, let $f_i^{w_i}(x)$ be some part of a NN $f^w(x)$, e.g., a layer or a subnetwork.

**Network morphism Type I.** We replace $f_i^{w_i}$ by

$$\tilde{f}_i^{\tilde{w}_i}(x) = A f_i^{w_i}(x) + b, \tag{2}$$

with $\tilde{w}_i = (w_i, A, b)$[1] . Equation (1) obviously holds for $A = \mathbf{1}, b = \mathbf{0}$. This morphism can be used to add a fully-connected or convolutional layer, as these layers are simply linear mappings. Chen et al. (2015) dubbed this morphism "Net2DeeperNet". Alternatively to the above replacement, one could also choose

$$\tilde{f}_i^{\tilde{w}_i}(x) = C(A f_i^{w_i}(x) + b) + d, \tag{3}$$

with $\tilde{w}_i = (w_i, C, d)$. $A, b$ are fixed, non-learnable. In this case network morphism Equation (1) holds if $C = A^{-1}, d = -Cb$. A Batch Normalization layer (or other normalization layers) can be written in the above form: $A, b$ represent the batch statistics and $C, d$ the learnable scaling and shifting.

**Network morphism Type II.** Assume $f_i^{w_i}$ has the form $f_i^{w_i}(x) = A h^{w_h}(x) + b$ for an arbitrary function $h$. We replace $f_i^{w_i}$, $w_i = (w_h, A, b)$, by

$$\tilde{f}_i^{\tilde{w}_i}(x) = \begin{pmatrix} A & \tilde{A} \end{pmatrix} \begin{pmatrix} h^{w_h}(x) \\ \tilde{h}^{w_{\tilde{h}}}(x) \end{pmatrix} + b \tag{4}$$

with an arbitrary function $\tilde{h}^{w_{\tilde{h}}}(x)$. The new parameters are $\tilde{w}_i = (w_i, w_{\tilde{h}}, \tilde{A})$. Again, Equation (1) can trivially be satisfied by setting $\tilde{A} = 0$. We think of two modifications of a NN which can be expressed by this morphism. Firstly, a layer can be widened (i.e., increasing the number of units in a fully connected layer or the number of channels in a CNN - the Net2WiderNet transformation from Chen et al. (2015)). Think of $h(x)$ as the layer to be widened. For example, we can then set $\tilde{h} = h$ to simply double the width. Secondly, skip-connections by concatenation as used by Huang et al. (2016) can be formulated as a network morphism. If $h(x)$ itself is a sequence of layers, $h(x) = h_n(x) \circ \cdots \circ h_0(x)$, then one could choose $\tilde{h}(x) = x$ to realize a skip from $h_0$ to the layer subsequent to $h_n$.

**Network morphism Type III.** By definition, every idempotent function $f_i^{w_i}$ can simply be replaced by

$$f_i^{(w_i, \tilde{w}_i)} = f_i^{\tilde{w}_i} \circ f_i^{w_i} \tag{5}$$

with the initialization $\tilde{w}_i = w_i$. This trivially also holds for idempotent function without weights, e.g., Relu.

**Network morphism Type IV.** Every layer $f_i^{w_i}$ is replaceable by

$$\tilde{f}_i^{\tilde{w}_i}(x) = \lambda f_i^{w_i}(x) + (1 - \lambda) h^{w_h}(x), \quad \tilde{w}_i = (w_i, \lambda, w_h) \tag{6}$$

with an arbitrary function $h$ and Equation (1) holds if $\lambda$ is initialized as 1. This morphism can be used to incorporate any function, especially any non-linearities. For example, Wei et al. (2016) use a special case of this operator to deal with non-linear, non-idempotent activation functions. Another example would be the insertion of an additive skip connection, which were proposed by He et al. (2016) to simplify training: If $f_i^{w_i}$ itself is a sequence of layers, $f_i^{w_i} = f_{i_n}^{w_{i_n}} \circ \cdots \circ f_{i_0}^{w_{i_0}}$, then one could choose $h(x) = x$ to realize a skip from $f_{i_0}^{w_{i_0}}$ to the layer subsequent to $f_{i_n}^{w_{i_n}}$.

Note that every combinations of the network morphisms again yields a morphism. So one could for example insert a block "Conv-BatchNorm-Relu" subsequent to a Relu layer by using equations (2), (3) and (5).

---

[1]In abuse of notation.

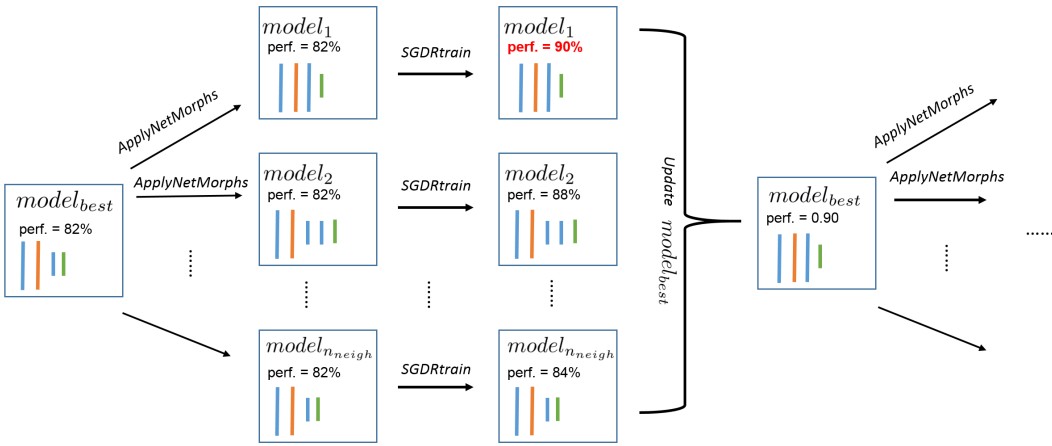

Figure 1: Visualization of our method. Based on the current best model, new models are generated and trained afterwards. The best model is than updated.

## 4 ARCHITECTURE SEARCH BY NETWORK MORPHISMS

Our proposed algorithm is a simple hill climbing strategy (Russell & Norvig, 2009). We start with a small, (possibly) pretrained network. Then, we apply network morphisms to this initial network to generate larger ones that may perform better when trained further. These new "child" networks can be seen as neighbors of the initial "parent" network in the space of network architectures. Due to the network morphism Equation (1), the child networks start at the same performance as their parent. In essence, network morphisms can thus be seen as a way to initialize child networks to perform well, avoiding the expensive step of training them from scratch and thereby reducing the cost of their evaluation. The various child networks can then be trained further for a brief period of time to exploit the additional capacity obtained by the network morphism, and the search can move on to the best resulting child network. This constitutes one step of our proposed algorithm, which we dub Neural Architecture Search by Hill-climbing (NASH). NASH can execute this step several times until performance on a validation set saturates; we note that this greedy process may in principle get stuck in a poorly-performing region, from which it can never escape, but we did not find evidence for this in our experiments.

Figure 1 visualizes one step of the NASH approach, and Algorithm 1 provides full details for the algorithm. In our implementation, the function $ApplyNetMorph(model, n)$ (line 15) applies $n$ network morphisms, each of them sampled uniformly at random from the following three:

- Make the network deeper, i.e., add a "Conv-BatchNorm-Relu" block as described at the end of Section 3. The position where to add the block, as well as the kernel size ($\in \{3, 5\}$), are uniformly sampled. The number of channels is chosen to be equal to he number of channels of the closest preceding convolution.

- Make the network wider, i.e., increase the number of channels by using the network morphism type II. The conv layer to be widened, as well as the widening factor ($\in \{2, 4\}$) are sampled uniformly at random.

- Add a skip connection from layer i to layer j (either by concatenation or addition – uniformly sampled) by using network morphism type II or IV, respectively. Layers i and j are also sampled uniformly.

Note that the current best model is also considered as a child, i.e. our algorithm is not forced to select a new model but can rather also keep the old one if no other one improves upon it.

---

**Algorithm 1** Network architecture search by hill climbing

1: **function** NASH( $model_0, n_{steps}, n_{neigh}, n_{NM}, epoch_{neigh}, epoch_{final}, \lambda_{end}, \lambda_{start}$ )
2:
3:      # $model_0 \triangleq$ model to start with,     $n_{steps} \triangleq$ number of hill climbining steps
4:      # $n_{neigh} \triangleq$ number of neighbours,     $n_{NM} \triangleq$ number of net. morph. applied
5:      # $epoch_{neigh} \triangleq$ number of epochs for training every neighbour
6:      # $epoch_{final} \triangleq$ number of epochs for final training
7:      # initial LR $\lambda_{start}$ is annealed to $\lambda_{end}$ during SGDR training
8:
9:      $model_{best} \leftarrow model_0$
10:      # start hill climbing
11:      **for** $i \leftarrow 1, \ldots, n_{steps}$ **do**
12:          #get $n_{neigh}$ neighbors of $model_0$ by applying $n_{NM}$ network morphisms to $model_{best}$
13:          **for** $j \leftarrow 1, \ldots, n_{neigh} - 1$ **do**
14:              $model_j \leftarrow ApplyNetMorphs(model_{best}, n_{NM})$
15:              # train for a few epochs on training set with SGDR
16:              $model_j \leftarrow$ SGDRtrain$(model_j, epoch_{neigh}, \lambda_{start}, \lambda_{end})$
17:          **end for**
18:          # in fact, last neighbor is always just the current best
19:          $model_{n_{neigh}} \leftarrow$ SGDRtrain$(model_{best}, epoch_{neigh}, \lambda_{start}, \lambda_{end})$
20:          # get best model on validation set
21:          $model_{best} \leftarrow \underset{j=1,\ldots,n_{neigh}}{arg\,max} \{performance_{vali}(model_j)\}$
22:      **end for**
23:      # train the final model on training and validation set
24:      $model_{best} \leftarrow$ SGDRtrain$(model_{best}, epoch_{final}, \lambda_{start}, \lambda_{end})$
25:      **return** $model_{best}$
26: **end function**

---

It is important for our method that child networks only need to be trained for a few epochs[2] (line 17). Hence, an optimization algorithm with good anytime performance is required. Therefore, we chose the cosine annealing strategy from Loshchilov & Hutter (2017), whereas the learning rate is implicitly restarted: the training in line 17 always starts with a learning rate $\lambda_{start}$ which is annealed to $\lambda_{end}$ after $epoch_{neigh}$ epochs. We use the same learning rate scheduler in the final training (aside from a different number of epochs).

While we presented our method as a simple hill-climbing method, we note that it can also be interpreted as a very simple evolutionary algorithm with a population size of $n_{neigh}$, no cross-over, network morphisms as mutations, and a selection mechanism that only considers the best-performing population member as the parent for the next generation. This interpretation also suggests several promising possibilities for extending our simple method.

## 5 EXPERIMENTS

We evaluate our method on CIFAR-10 and CIFAR-100. First, we investigate whether our considerations from the previous chapter coincide with empirical results. We also check if the interplay of modifying and training networks harms their eventual performance. Finally, we compare our proposed method with other automated architecture algorithms as well as hand crafted architectures.

We use the same standard data augmentation scheme for both CIFAR datasets used by Loshchilov & Hutter (2017) in all of the following experiments. The training set (50.000 samples) is split up in training (40.000) and validation (10.000) set for the purpose of architecture search. Eventually the performance is evaluated on the test set. All experiments where run on Nvidia Titan X (Maxwell)

---

[2]I.e., $epoch_{neigh}$ should be small since a lot of networks need to be trained. In fact, the total number of epochs for training in our algorithm can be computed as $epoch_{total} = epoch_{neigh} n_{neigh} n_{steps} + epoch_{final}$. In our later experiments we chose $epoch_{neigh} = 17$.

GPUs, with code implemented in Keras (Chollet et al., 2015) with a TensorFlow (Abadi et al., 2015) backend.

## 5.1 EXPERIMENTS ON CIFAR-10

### 5.1.1 BASELINES

Before comparing our method to others, we run some baseline experiments to see whether our considerations from the previous chapter coincide with empirical data.

**Random model selection.** First, we investigate if the simple hill climbing strategy is able to distinguish between models with high and low performance. For this, we set $n_{neigh} = 1$, i.e., there is no model selction - we simply construct random networks and train them. We then run experiments with $n_{neigh} = 8$ and compare both results. All other parameters are the same in this experiment, namely $n_{steps} = 5, n_{NM} = 5, epoch_{neigh} = 17, epoch_{final} = 100$. We choose $\lambda_{start} = 0.05, \lambda_{end} = 0.0$ as done in Loshchilov & Hutter (2017). $model_0$ was a simple conv net: Conv-MaxPool-Conv-MaxPool-Conv-FC-Softmax[3], which is pretrained for 20 epochs, achieving $\approx 75\%$ validation accuracy (up to $91\%$ when trained till convergence), see Figure 5 in the appendix. If our algorithm is able to identify better networks, one would expect to get better results with the setting $n_{neigh} = 8$.

**Retraining from scratch.** In the this experiment we investigate whether the "weight inheritance" due to the network morphisms used in our algorithm harms the final performance of the final model. This weight inheritance can be seen as a strong prior on the weights and one could suspect that the new, larger model may not be able to overcome a possibly poor prior. Additionally we were interested in measuring the overhead of the architecture search process, so we compared the times for generating and training a model with the time needed when training the final model from scratch. The retraining from scratch is done for the same number of epochs as the total number of epochs spent to train the model returned by our algorithm[4] .

**No SGDR.** We now turn off the cosine annealing with restarts (SGDR) during the hill climbing stage, i.e., the training in line 17 of Algorithm 1 is done with a constant learning rate. We tried $\lambda \in \{0.01, 0.025, 0.05\}$, 10 runs each and averaged the results. Note that we still use the cosine decay for the final training.

**No network morphism.** Lastly, we turn off the network morphism constraint for initializing the neighbor networks. In detail, we proceeded as Real et al. (2017): All weights from layer where now changes occur are inherited, whereas the weights of new/modified layers are initialized by random.

The results for these experiments are summarized in Table 1 . The hill climbing strategy is actually able to identify better performing models. (first and second line: $5.7\%$ vs. $6.5\%$). Notice how hill climbing prefers larger models ($5.7$ million parameters on average vs. $4.4$ million). Performance slightly decreases when the models are retrained from scratch (line 3). This experiments indicates that our algorithm does not harm the final performance of a model. Regarding the runtime, the overhead for first having to search for the architecture is roughly a factor 3. We think this is a big advantage of our method and shows that architecture search can be done in the same order of magnitude as training a single model. In line 4 we can see that SGDR plays an important role. The resulting models chosen by our algorithm when training is done with a constant learning rate perform similarly to the models without any model selection strategy ($6.4\%$ and $6.5\%$, respectively), which indicates that the performance after a few epochs on the validation set when trained without SGDR correlates less with the final performance on the test set as it is the case for training with SGDR. Indeed, we computed the Pearson correlation coefficient and obtained $R^2 = 0.64$ for training with SGDR and and $R^2 = 0.36$ for training with a constant learning rate. See appendix A. Also, with the constant learning rate, the few epochs spent are not sufficient to improve the performance of the model. Figure 2 shows the progress while running our algorithm with and without SGDR, averaged over all runs. When turning off the network morphism constraint, performance also decreases.

---

[3]By Conv we actually mean Conv-BatchNorm-Relu.

[4]With the setting $n_{steps} = 5, epoch_{neigh} = 17, epoch_{final} = 100$ and pretraining the starting network for 20 epochs, the models that are returned by our algorithm are trained for a total number of of $20 + 17 \cdot 5 + 100 = 205$ epochs.

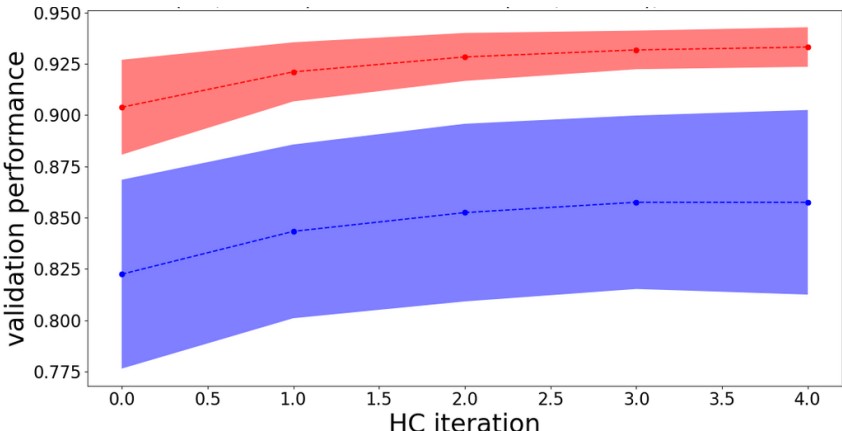

Figure 2: The best model found by Algorithm 1 tracked over time (in terms of hill climbing iterations). With (red) and without (blue) using SGDR for the training within the hill climbing (line 17). Final training (line 24) is not plotted. Dashed line denotes mean, shaded area $\pm 2\sigma$ intervalls.

Table 1: Baseline experiments. Runtime, # params, and error rates are averaged over 10 runs (for $n_{neigh} = 8$) and 30 runs ($n_{neigh} = 1$) runs, respectively. $n_{steps} = 5$ in all experiments.

| algorithm setting | runtime (hrs) | # params (mil.) | error $\pm$ std. ( %) |
|---|---|---|---|
| $n_{neigh} = 8$ | 12.8 | 5.7 | $5.7 \pm 0.35$ |
| Random networks $(n_{neigh} = 1)$ | 4.5 | 4.4 | $6.5 \pm 0.76$ |
| models from line 1 retrained from scratch | 5.3 | 5.7 | $6.1 \pm 0.92$ |
| $n_{neigh} = 8$, no SGDR | 10.6 | 5.8 | $6.4 \pm 0.70$ |
| $n_{neigh} = 8$, no net. morph. | 6.6 | 2.9 | $6.1 \pm 0.30$ |

Interestingly the number of parameters heavily decreases. This indicates that our algorithm prefers models without new parameters.

### 5.1.2 COMPARISON TO HAND CRAFTED AND OTHER AUTOMATICALLY GENERATED ARCHITECTURES

We now compare our algorithm against the popular wide residual networks (Zagoruyko & Komodakis, 2016), the state of the art model from Gastaldi (2017) as well as other automated architecture search methods. Beside our results for $n_{steps} = 5$ from the previous section, we also tried $n_{steps} = 8$ to generate larger models.

For further improving the results, we take snapshots of the best models from every iteration while running our algorithm following the idea of Huang et al. (2017) when using SGDR (Loshchilov & Hutter, 2017) for training. However different from Huang et al. (2017), we do not immediately get fully trained models for free, as our snapshots are not yet trained on the validation set but rather only on the training set. Hence we spent some additional resources and train the snapshots on both sets. Afterwards the ensemble model is build by combining the snapshot models with uniform weights. Lastly, we also build an ensemble from the models returned by our algorithm across all runs. Results are listed in Table 2.

The proposed method is able to generate competitive network architectures in only 12 hours. By spending another 12 hours, it outperforms most automated architecture search methods although all of them require (partially far) more time and GPUs. We do not reach the performance of the two handcrafted architectures as well as the ones found by Zoph & Le (2017) and Brock et al. (2017). However note that Zoph & Le (2017) spent by far more resources than we did.

Unsurprisingly, the ensemble models perform better. It is a simple and cheap way to improve results which everyone can consider when the number of parameters is not relevant.

Table 2: Results for CIFAR-10. For our methods the stated resources, # parameters and errors are averaged over all runs. "Resources spent" denotes training costs in case of the handcrafted models.

| model | resources spent | # params (mil.) | error ( %) |
|---|---|---|---|
| Shake-Shake (Gastaldi, 2017) | 4 GPU days, 2 GPUs | 26 | 2.9 |
| WRN 28-10 (Loshchilov & Hutter, 2017) | 1 GPU day | 36.5 | 3.86 |
| Baker et al. (2016) | 80-100 GPU days | 11 | 6.9 |
| Cai et al. (2017) | 15 GPU days | 19.7 | 5.7 |
| Zoph & Le (2017) | 16.000-24.000 GPU days | 37.5 | 3.65 |
| Real et al. (2017) | 2500 GPU days | 5.4 | 5.4 |
| Saxena & Verbeek (2016) | ? | 21 | 7.4 |
| Brock et al. (2017) | 3 GPU days | 16.0 | 4.0 |
| Ours (random networks, $n_{steps} = 5, n_{neigh} = 1$) | 0.2 GPU days | 4.4 | 6.5 |
| Ours ($n_{steps} = 5, n_{neigh} = 8$, 10 runs) | 0.5 GPU days | 5.7 | 5.7 |
| Ours ($n_{steps} = 8, n_{neigh} = 8$, 4 runs) | 1 GPU day | 19.7 | 5.2 |
| Ours (snapshot ensemble, 4 runs) | 2 GPU days | 57.8 | 4.7 |
| Ours (ensemble across runs) | 4 GPU days | 88 | 4.4 |

Table 3: Results for CIFAR-100. For our methods the stated resources, # parameters and errors are averaged over all runs. "Resources spent" denotes training costs in case of the handcrafted models.

| model | resources spent | # params (mil.) | error ( %) |
|---|---|---|---|
| Shake-Shake (Gastaldi, 2017) | 14 GPU days | 34.4 | 15.9 |
| WRN 28-10 (Loshchilov & Hutter, 2017) | 1 GPU day | 36.5 | 19.6 |
| Real et al. (2017) | 250 GPUs | 40.4 | 23.7 |
| Brock et al. (2017) | 3 GPU days | 16.0 | 20.6 |
| Ours ($n_{steps} = 8, n_{neigh} = 8$, 5 runs) | 1 GPU day | 22.3 | 23.4 |
| Ours (snapshot ensemble, 5 runs) | 2 GPU days | 73.3 | 20.9 |
| Ours (ensemble across runs) | 5 GPU days | 111.5 | 19.6 |

## 5.2 EXPERIMENTS ON CIFAR-100

We repeat the previous experiment on CIFAR-100; hyperparameters were not changed. The results are listed in Table 3. Unfortunately most automated architecture methods did not consider CIFAR-100. Our method is on a par with Real et al. (2017) after one day with a single GPU. The snapshot ensemble performs similar to Brock et al. (2017) and an ensemble model build from the 5 runs can compete with the hand crafted WRN 28-10. The performance of the Shake-Shake network (Gastaldi, 2017) is again not reached.

## 6 CONCLUSION

We proposed NASH, a simple and fast method for automated architecture search based on a hill climbing strategy, network morphisms, and training via SGDR. Experiments on CIFAR-10 and CIFAR-100 showed that our method yields competitive results while requiring considerably less computational resources than most alternative approaches. Our algorithm is easily extendable, e.g., by other network morphisms, evolutionary approaches for generating new models, other methods for cheap performance evaluation (such as, e.g., learning curve prediction (Klein et al., 2017) or hypernetworks (Ha et al., 2017; Brock et al., 2017)), or better resource handling strategies (such as Hyperband (Li et al., 2016b)). In this sense, we hope that our approach can serve as a basis for the development of more sophisticated methods that yield further improvements of performance.

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

# A  CORRELATION BETWEEN VALIDATION AND TEST ACCURACY

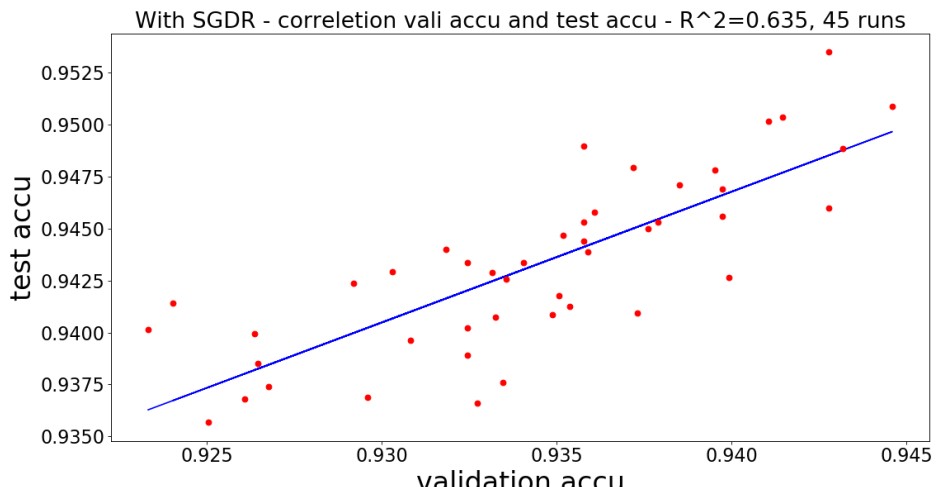

Figure 3: Initial network for our algorithm.

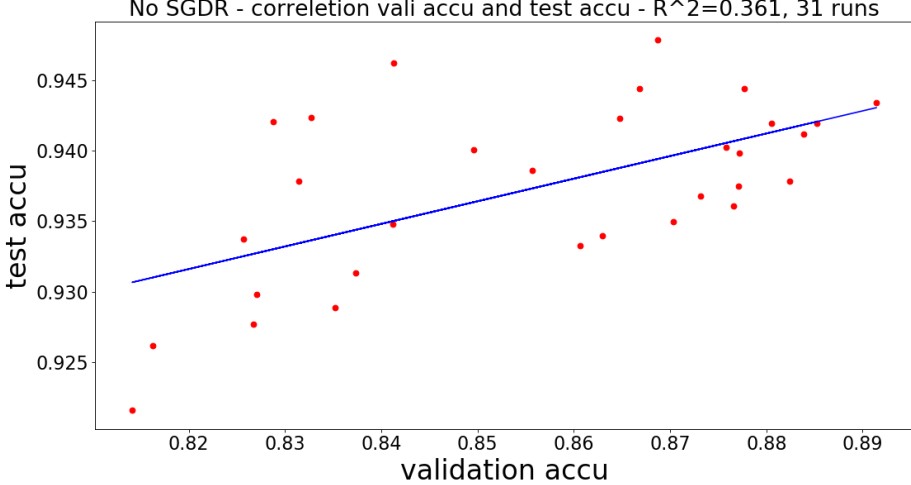

Figure 4: Initial network for our algorithm.

# B  SOME MODELS

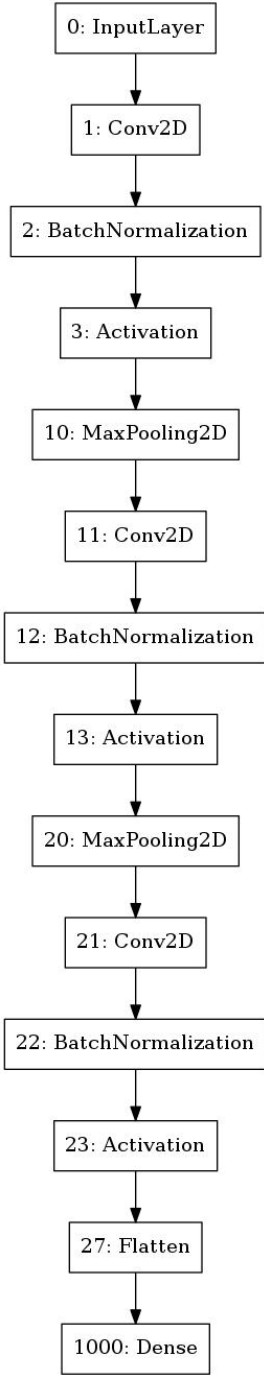

Figure 5: Initial network for our algorithm.

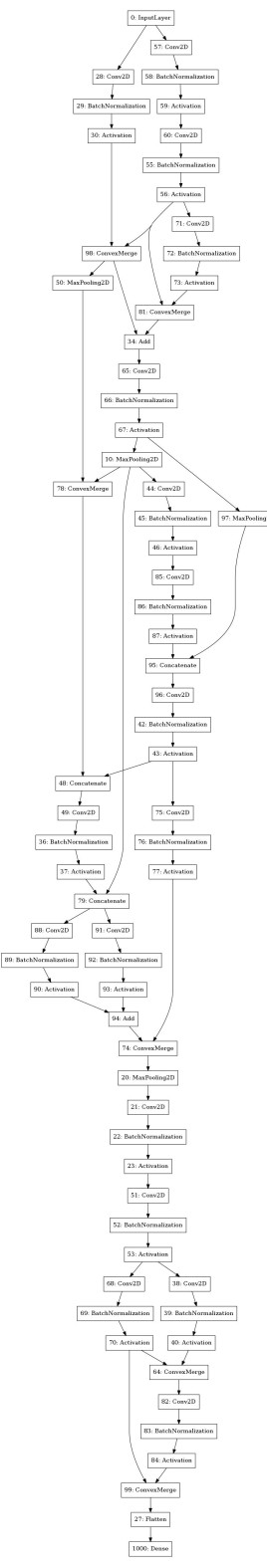

Figure 6: Network generated by our algorithm with $n_{steps} = 5$.

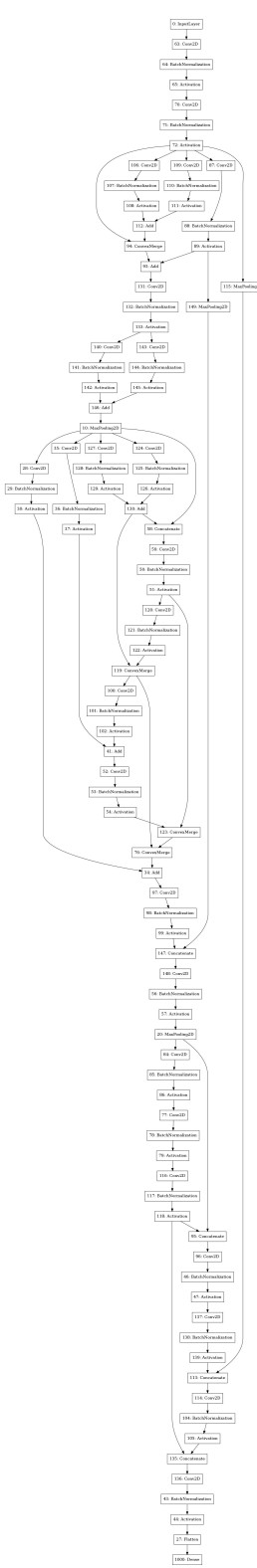

Figure 7: Network generated by our algorithm with $n_{steps} = 8$.

