# OpenReview forum: "Simple and efficient architecture search for Convolutional Neural Networks"
_ICLR.cc/2018/Conference — Invite to Workshop Track_

### Official Review · AnonReviewer2 · 2017-11-25
**Impressive results, not very fair comparison.**

**Rating:** 6
**Confidence:** 4

**Review:**

This paper proposes a neural architecture search method that achieves close to state-of-the-art accuracy on CIFAR10 and takes much less computational resources. The high-level idea is similar to the evolution method of [Real et al. 2017], but the mutation preserves net2net properties, which means the mutated network does not need to retrain from scratch.

Compared to other papers on neural architecture search, the required computational resource is impressively small: close to state-of-the-art result in one day on a single GPU. However, it is not clear to me what contribute to the massive improvement of speed. Is it due to the network morphing that preserve equality? Is it due to a good initial network structure? Is it due to the well designed mutation operations? Is it due to the simple hill climbing procedure (basically evolution that only preserve the elite)? Is it due to a well crafted search space that is potentially easier?

The experiments in this paper does not provide enough evidence to tease apart the possible causes of this dramatic reduction on computational resources. And the comparisons to other papers seems not fair since they all operate on different search space.

In summary, getting net2net to work for architecture search is interesting. And I love the results. These are very impressive numbers for neural architecture search. However, I am not convinced that the improve is resulted from a better algorithm. I would suggest that the paper carefully evaluates each component of the algorithm and understand why the proposed method takes far less computational resources.

---

> ### Author Response · Authors · 2017-12-31
> **Just some comments on the feedback.**
>
> Thank you for the comment.
> Regarding your concern what actually yields the massive speed improvement: We will update the paper in just a few days, where you will find an additional experiment: We run our algorithm without the network morphism constraint but rather only ‘inherit’ unchanged weights (this is in essence what Real et al. in Large-Scale Evolution of Image Classifiers did).  You will then find in section 5.1.1 (baseline experiments on CIFAR10) that performance deteriorates if we either 1) turn model selection off or 2) turn SGDR off or 3) turn the network morphism constraint off. In conclusion, all three aspects of the method are essential and contribute to the speed up.

---

### Official Review · AnonReviewer3 · 2017-11-27
**Good but with limitations.**

**Rating:** 5
**Confidence:** 5

**Review:**

This paper presents a method to search neural network architectures at the same time of training. It does not require training from scratch for each architecture, thus dramatically saves the training time. The paper can be understood with no problem.  Moderate novelty, network morphism is not novel, applying it to architecture search is novel.

Pros:
1. The required time for architecture searching is significantly reduced.
2. With the same number or less of parameters, this method is able to outperform previous methods, with much less time.

However, the method described is restricted in the following aspects.

1. The accuracy of the training set is guaranteed to ascend because network morphism is smooth and number of params is always increasing, this also makes the search greedy , which could be suboptimal. In addition, the algorithm in this paper selects the best performing network at each step, which also hampers the discover of the optimal model.

2. Strong human prior, network morphism IV is more general than skip connection, for example, a two column structure belongs to type IV. However, in the implementation, it is restricted to skip connection by addition. This choice could be motivated from the success of residual networks. This limits the method from discovering meaningful structures. For example, it is difficult to discover residual network denovo. This is a common problem of architecture searching methods compared to handcrafted structures.

3. The comparison with Zoph & Le is not fair because their controller is a meta-network and the training happens only once. For example, the RNNCell discovered can be fixed and used in other tasks, and the RNN controller for CNN architecture search could potentially be applied to other tasks too (though not reported).

---

> ### Author Response · Authors · 2017-12-31
> **Just some comments on the feedback.**
>
> Thank you for the comment.
> Regarding 1. and 2.: Yes, we agree with that. We will work on this.
> Regarding 3.: Even though we searched for the whole architecture, our method could in principle also be used to learn cells/blocks, which could then be reused for other problems.

---

### Official Review · AnonReviewer1 · 2017-11-27
**review: questionable utility**

**Rating:** 4
**Confidence:** 4

**Review:**

This paper proposes a variant of neural architecture search.  It uses established work on network morphisms as a basis for defining a search space.  Experiments search for effective CNN architectures for the CIFAR image classification task.

Positives:

(1) The approach is straightforward to implement and trains networks in a reasonable amount of time.

(2) An advantage over prior work, this approach integrates architectural evolution with the training procedure.  Networks are incrementally grown; child networks are initialized with learned parameters from their parents.  This eliminates the need to restart training when making an architectural change, and drastically speeds the search.

Negatives:

(1) The state-of-the-art CNN architectures are not mysterious or difficult to find, despite the paper's characterization of them being so.  Indeed, ResNet and DenseNet designs are both guided by extremely simple principles: stack a series of convolutional layers, pool occasionally, and use some form of skip-connection throughout.  The need for architectural search is unclear.

(2) The proposed search space is boring.  As described in Section 4, the possibly evolutionary changes are limited to deepening the network, widening the network, and adding a skip connection.  But these are precisely the design aspects that have been well-explored by human trial and error and for which good rules of thumb are already available.

(3) As a consequence of (1) and (2), the result is essentially rigged.  Since only depth, width, and skip connections are considered, the end network must end up looking like a ResNet or DenseNet, but with some connections pruned.  There is no way to discover a network outside of the principled design space articulated in point (1) above.  Indeed, the discovered network diagrams (Figures 4 and 5) fall in this space.

(4) Performance is worse than the best hand-designed baselines.  One would hope that, even if the search space is limited, the discovered networks might be more efficient or higher performing in comparison to the human designs which fall within that same space.  However, the results in Tables 3 and 4 show this not to be the case.  The best human designs outperform the evolved networks.  Moreover, the evolved networks are woefully inefficient in terms of parameter count.

Together, these negatives imply the proposed approach is not yet at the point of being useful in practice.  I think further work is required (perhaps expanding the search space) to resolve the current limitations of automated architecture search.

Misc:

Tables 3 and 4 would be easier to parse if resources were simply reported in terms of total GPU hours.

---

> ### Author Response · Authors · 2017-12-31
> **Just some comments on the feedback.**
>
> Thank you for the comment.
> Regarding your negatives:
> (1)	We agree with the simple design principles pointed out. However, it took quit a long time till people came up with them. Beside that, these principles depend on the input and target domain. There are certainly other, not well understood, domains, were such principles do not exist, and ideally (at some point in the future) an architectures search algorithm can help in these situations. We agree that there may not be need for architecture search to improve CIFAR image classification (because it is already well understood and basically ‘solved’), but our paper presents a general method applicable to arbitrary domains. The reason why we chose CIFAR is a) a simple data set to start with, b) benchmarking and c) computational constraints.
> (2)	We would like to note that we give just a few examples of network morphisms – they can certainly be extended by other, not so boring, ones.
> (4)	Yes, we do not reach hand-designed baselines. However note that even training some of these networks (e.g., shake shake network) takes longer than our architecture search algorithm. Also hyperparameters and the training procedure are often highly optimized for state of the art hand crafted architectures, which is not the case for our networks.

---

### Author Response · Authors · 2018-01-03
**Update**

Dear readers,

we updated the experimental section of our paper.

---

### Decision · Program_Chairs · 2018-01-29
**ICLR 2018 Conference Acceptance Decision**

**Decision:**

Invite to Workshop Track

**Comment:**

The paper proposes a method for architecture search using network morphisms, which allows for faster search without retraining candidate models. The results on CIFAR are worse than the state of the art, but reasonably competitive, and achieved using limited computation resources. It would have been interesting to see how the method would perform on large datasets (ImageNet) and/or other tasks and search spaces. I would encourage the authors to extend the paper with further experimental evaluation.